# Frequency of *Candida* spp. in the Oral Cavity of Asymptomatic Preschool Mexican Children and Its Association with Nutritional Status

**DOI:** 10.3390/children9101510

**Published:** 2022-10-03

**Authors:** Rodolfo Pinto-Almazán, María Guadalupe Frías-De-León, Claudia Erika Fuentes-Venado, Roberto Arenas, Leopoldo González-Gutiérrez, Edwin Chávez-Gutiérrez, Oscar Uriel Torres-Paez, Erick Martínez-Herrera

**Affiliations:** 1Sección de Estudios de Posgrado e Investigación, Escuela Superior de Medicina, Instituto Politécnico Nacional, Plan de San Luis y Díaz Mirón, Ciudad de México 11340, Mexico; 2Unidad de Investigación, Hospital Regional de Alta Especialidad de Ixtapaluca, Ixtapaluca 56530, Mexico; 3Servicio de Medicina Física y Rehabilitación, Hospital General de Zona No. 197, Texcoco 56108, Mexico; 4Sección de Micología, Hospital General “Dr. Manuel Gea González”, Ciudad de México 14080, Mexico; 5Clínica Sanvite, Guadalajara 44110, Mexico; 6Doctorado en Biomedicina y Biotecnología Molecular, Nacional de Ciencias Biológicas, Instituto Politécnico Nacional, Av. Luis Enrique Erro S/N, Unidad Profesional Adolfo López Mateos, Zacatenco, Alcaldía Gustavo A. Madero, Ciudad de México 07738, Mexico

**Keywords:** *Candida* spp., children, oral colonization, nutritional status, Mexico

## Abstract

Malnutrition is a public health problem in developing countries, affecting the child population, which favors the appearance of infections such as oral candidiasis. In Mexico, information on the presence of oral colonization by *Candida* spp. in asymptomatic children is scarce. The present study aimed to determine the presence of *Candida* spp. in the oral cavity of asymptomatic preschool Mexican children and its association with their nutritional status. A sample of oral mucosa was obtained using a sterile swab and then inoculated in Sabouraud dextrose agar with antibiotics, and the yeast growth was phenotypically identified. The anthropometric profile of children was performed based on the guidelines of the International Society for the Advancement of Kinanthropometry. In addition, eating habits were investigated. The possible associations between the variables were determined through the chi-square test (IC95%, *p* < 0.05) (GraphPad Prism 8.0). Among the 743 assessed children (403 boys and 340 girls), the average age was 4.6 years, and the average nutritional status was normal (53.7%), followed by undernutrition (28.4%), overweight (12.4%) and obesity (5.5%). In 52 children, *Candida* was isolated, and the identified species were *C. albicans* (87.8%), *C. glabrata* (11.5%), *C. krusei* (5.8%) and *C. parapsilosis* (1.9%). The frequency of colonization was greater in males of six years (69.23%). There was no significant association between the colonization by *Candida* spp. and the nutritional status; however, a relation was observed with a high intake of simple carbohydrates.

## 1. Introduction

The genus *Candida* is composed of more than 200 yeast species, some of which are found as commensals in the gastrointestinal tract, periorificial skin and mucous membranes (vaginal and oral); however, nearly 10% of the species may behave as opportunistic pathogens, causing infections such as oral candidiasis [1].

The conversion from commensal to pathogen depends on various factors, both local and systemic, among which are the immature immune system in children under 4 years of age, pregnancy, immunosuppression associated with infection with the human immunodeficiency virus (HIV), cancer, chronic diseases (diabetes mellitus), and treatment with broad-spectrum antibiotics, as well as nutritional factors [2,3]. It has been demonstrated that nutritional deficiencies that are involved in the pathogenesis of oral candidosis include deficiencies of iron, folate, vitamin B12, C and possibly A, as well as protein-energy malnutrition [2]. These deficiencies reduce host resistance and cause the loss of epithelial integrity of the oral mucosa, which facilitates the invasion of the hyphae to establish infection. On the contrary, excessive intake of carbohydrates has been shown to increase the incidence of oral infection by *Candida*. It has been suggested that such diets are likely to facilitate the adherence of *Candida* to the epithelial cells of the oral mucosa [2].

Malnutrition is a worldwide public health problem worldwide, particularly in developing countries, such as Mexico, where the child population is the most susceptible, favoring the appearance of a variety of infections, including thrush in the mouth. The relationship between malnutrition and infection is a vicious circle, where poor nutritional factors such as malnutrition can make a child more susceptible to serious and recurrent infections, and these in turn contribute to the development of malnutrition [4]. 

Among the species of *Candida* that are isolated with greater frequency in oral candidiasis are: *Candida albicans*, *C. dubliniensis*, *C. glabrata*, *C. tropicalis*, *C. krusei*, *C. parapsilosis*, *C. kefyr* and *C. famata*. These species have been isolated particularly from child and adult populations with HIV/AIDS and malnutrition [5,6,7,8].

Previous reports have found that in order to enforce preventive programs for children, it is necessary to know the prevalence and which Candida species are potentially pathogenic in the oral cavity of undernourished children. However, information about the frequency of Candida in the pediatric population with nutritional alterations is scarce.

Therefore, the aim of the present study was to determine the presence of *Candida* spp. in the oral cavity of preschool asymptomatic Mexican children and its association with their nutritional status.

## 2. Materials and Methods

An observational, descriptive, prospective and cross-sectional study was carried out in a group of 981 preschool children from the Municipality of Chiconcuac, located in the northeastern part of the State of Mexico, Mexico, which is a socially marginalized region.

The inclusion criteria were: (1) children; (2) asymptomatic for oral infection by *Candida*; (3) asymptomatic for infection of the throat; (4) not having received antifungal treatment during the last three months; and (5) not using mouth rinses or tooth brushing for at least eight hours prior to sampling.

Before to the completion of the study, informed consent was obtained from parents or guardians. 

### 2.1. Anthropometry

An anthropometric analysis of the children was performed, taking into account the following variables: weight (kg), height (cm), skin folds (bicipital, tricipital, subscapular, and suprailiac regions (mm)), mean circumference of arm (cm) and head circumference (cm), in accordance with the guidelines for the anthropometric evaluation published by the International Society for the Advancement of Kinanthropometry (ISAK). According to the above variables, we estimated the arm muscle area, the percentage of fat mass and body mass index for age (BMIA). The Z scores, percentiles, weight-for-age, height-for-age and weight-for-height and BMIA were calculated using the WHO Anthro programs (version 3.2.2, January 2011) and WHO Anthro Plus (version 3.2.2, 2007).

Furthermore, parents or guardians were asked about the children’s eating habits.

### 2.2. Sample and Cultivation

Each child took a sample of buccal mucosa (cheeks, the dorsum of the tongue, floor of the mouth and palate) with a sterile swab. The swabs were placed in tubes containing Stuart transport medium (BBL Culture Swab; Collection and Transport System of Becton, Dickinson and Company, Franklin Lakes, NJ, USA) and were taken to the laboratory for cultivation. The samples were inoculated in Sabouraud dextrose agar plates/chloramphenicol (Dibico, Stamford, CT, USA) and incubated for 48 h at 36 °C. To confirm the presence of yeast, a fresh smear was performed. The fresh smear was performed by means of obtention with a micology handle from a fragment of the sample in Sabouraud agar and placing it over a film with a previous preparation with a drop of sterile saline solution. Afterwards, we covered the preparation with a glass slide and observed it through the microscope at 40×. Yeast colonies were selected and reinoculated in Sabouraud dextrose agar. For positive yeast cultures, the number of colonies was recorded.

### 2.3. Phenotypic Identification

Yeasts were reinoculated in BBL CHROMagar *Candida* Medium (Becton Dickinson and Company, NJ, USA) and incubated for 24–48 h at 37 °C. The yeasts were identified on the basis of the color of the colonies (*C. albicans*—light green to medium; *C. krusei*—pale pink, purple (harsh with extended edges, pale); *C. tropicalis*—dark blue to grayish blue, with a dark halo in agar; *C. glabrata*—varies from white, pink or purple; *C. parapsilosis*—white to pale pink) [9,10]

### 2.4. Statistical Analysis

The possible associations between the oral colonization by *Candida* spp. and the nutritional status, sex, age, as well as the species of *Candida* identified, were determined through the chi-square test (IC95%, *p* ≤ 0.05) using the statistical package GraphPad Prism 8.0.

## 3. Results

A total of 981 children were invited to participate in the study, of whom 743 agreed to participate. 

### 3.1. Demographic and Anthropometric Data 

Of the total population of 743 preschool children studied (403 boys and 340 girls), the nutritional status was normal in the majority of them (399 children, 53.7%), followed by undernutrition (211 children, 28.4%), overweight (92 children, 12.4%) and obesity (41 children, 5.5%). The age of the population studied was between 3 and 6 years, with an average of 4.6.

In the consumption of carbohydrate context, parents or guardians mentioned that preschool children with *Candida* spp. consumed simple carbohydrates between three and four times a week (sweets, sugar, honey, etc.).

### 3.2. Cultivation

Of the 743 samples taken from the buccal mucosa, in 52, yeast isolation was obtained. The number of CFUs per plate ranged from one to seven. Overall, 24 isolates were obtained from children with a normal nutritional state, 21 from children with undernutrition, six from overweight children and one from a child with obesity.

### 3.3. Phenotypic Identification

Based on the color of the colony developed in CHROMagar, it was determined that the species present in the oral mucosa of the preschool children were: *C. albicans* (42, 87.8%), *C. glabrata* (6, 11.5%), *C. krusei* (3, 5.8%), and *C. parapsilosis* (1, 1.9%) (Figure 1).

### 3.4. Statistical Analysis

The evaluation of the association between variables showed no significant differences between them, except for between the presence or absence of *Candida* spp. and sex and the presence or absence of *Candida* spp. and age (Table 1).

Although no significant differences were observed in the chi-square test among the nutritional status groups, regarding the presence of *Candida*, when calculating the odds ratio (OR), the risk of having oral *Candida* due to malnutrition (undernutrition, overweight or obesity) was estimated at OR = 1.173 (CI 95% 0.6674 to 2.062). Likewise, while studying whether having high and very high consumption of carbohydrates was a risk factor, we found an OR = 1.148 (CI 95% 0.5386 to 2.447) (Table 2).

With regard to the results obtained, it was observed that colonization by *Candida* spp. was higher in male preschoolers (69.23%) than female preschoolers (30.77%). In relation to age, six-year-old male preschoolers had a higher frequency of colonization (Table 3). 

## 4. Discussion

*Candida* is the most commonly isolated commensal yeast in the oral cavity, reported in up to 70% of healthy individuals [11]. Oral colonization of *Candida* spp. may lead to oral or disseminated infection, depending on factors associated with the microorganism or the host, such as malnutrition, immunosuppression, poor oral hygiene and alterations in the salivary flow [12,13]. The reduction in the defenses promotes colonization by *Candida* and possibly an opportunistic infection that produces a broad spectrum of manifestations in the oral mucosa [11,14,15]. In the pediatric population, both the immune system and the oral microbiota are still immature, which, along with the nutritional alterations (undernutrition, overweight, and obesity) that are present at a global level, constitute the most important etiologic factors in the pathogenesis of oral candidiasis. This demonstrates the importance of knowing the frequency of yeasts from the genus *Candida* in preschool children and its association with nutritional status (normal, undernutrition, overweight and obesity) in order to implement preventive measures for oral candidiasis in this population. 

Malnutrition is characterized by an imbalance, either due to deficiency or excess, in the intake of nutrients and/or energy. Malnutrition is a very important global public health problem, as it involves both extremes of obesity and micronutrient deficiency or excess present in almost all countries, especially those considered underdeveloped [16,17]. Techniques based on body mass index, skinfold thickness, and waist or hip circumference have been extensively used for nutritional status analysis. Additionally, these techniques have been suggested to satisfactorily predict cardiovascular risk. However, in children under 5 years of age, nutritional status cannot be determined using BMI alone. In this population, it should be evaluated clinically based on w/a and w/h percentiles and pediatric BMI [18,19]. In the case of the preschool population, children with low weight are considered to have anthropometric parameters with a Z-score below −2 and a percentile below 5. As for children with a eutrophic or normal weight, their anthropometric data oscillate between Z-scores of −1 and 1 and the percentiles 5 and 85. Likewise, in the case of overweight children, they belong to the percentiles between 85 and 95 and have a Z-score > 2, and children with obesity belong to a percentile of more than 95 and have a Z-score of 3 or more.

In this study, a frequency of colonization of 6.99% was found in children from 3 to 6 years. This frequency is lower than that reported in other studies, which indicate a prevalence of 11.8 to 50.0% in asymptomatic pediatric oral carriers [20]. This difference may be due to the methodology used for sample collection, since it is known that there are various techniques for collection (oral rinse, cultivation by imprint, total collection of saliva, swab, and biopsy), which have different sensitivities in the detection of *Candida* [11,21,22].

In the present study, *C. albicans* was found to be the most common species, followed by *C. glabrata*, *C. kusei* and *C. parapsilosis*. Contrary to these results, Zöllner and Jorge [23], reported that *C. albicans* was the primary agent found in the mouth of infants during breastfeeding and artificial feeding, followed by *C. parapsilosis*, *C. tropicalis* and *C. guilliermondii* [23]. Furthermore, it is known that the composition of the buccal microbiota varies between populations due to geographic or social differences [24]. 

Among the analyzed population, the most significant colonization was found in six-year-old males. This finding is consistent with other studies in which the distribution of *Candida* spp. between ages was higher in children aged from 6 to 12 years [20,25].

With regard to sex, our results differ from those reported in other studies [11,20,24]. This situation may be because the male sample was larger than the female sample.

The association between colonization and the nutritional status was not significant, although it is known that undernutrition is a risk factor for oral yeast [26,27]. This result may be due to the size of the sample, considering that when the proportion of carrier children within the population of undernourished and healthy preschool children was analyzed, we observed that 9.9% of undernourished children were carriers compared to 6.0% of the population with normal nutritional status.

Concerning high carbohydrate intake, there is some controversy regarding whether or not it can promote colonization by *Candida* [28,29]. However, in accordance with Felea et al. [29], we noted that the presence of *Candida* spp. in the oral cavity was associated with a high intake of simple carbohydrates (more than three times a week) in 72.4% of preschooler carriers.

## 5. Conclusions

Overall, 6.99 percent of the preschool children included in this study were carriers of *Candida* spp. in the oral cavity, with *C. albicans* being the most common species. Six-year-old males were the main carriers of yeast. There was no significant association between the presence of *Candida* spp. and nutritional status, but a correlation was observed with a high intake of simple carbohydrates.

## Figures and Tables

**Figure 1 children-09-01510-f001:**
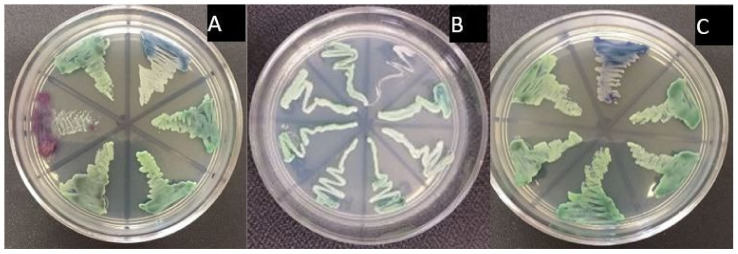
Phenotypic identification of *Candida* isolates in CHROMagar Candida. (**A**) *Candida albicans* (green), *Candida krusei* (pale pink), *Candida glabrata* (purple). (**B**) *Candida albicans* (green). (**C**) *Candida albicans* (green), *Candida glabrata* (purple).

**Table 1 children-09-01510-t001:** Comparison between the studied variables.

Association	Chi Square	*p*
The presence or absence of *Candida* spp. and individual nutritional status	4584	0.2049
The presence or absence of *Candida* spp. and sex	5063	0.0244 *
Species of *Candida* and sex	1207	0.7513
The presence or absence of *Candida* spp. and age	7941	0.0473 *
Species of *Candida* and age	7796	0.2534

* A probability value of *p* ≤ 0.05 was considered to indicate statistical significance.

**Table 2 children-09-01510-t002:** Risk factors for the presence of oral *Candida*.

	Cases (*n* = 52)		Controls (*n* = 691)		OR	IC 95%
	N	%	N	%		
Nutrition Eutrophic	26	50	373	53.98	1	
Malnutrition (Undernutrition, overweight and obesity)	26	50	318	46.02	1.173	0.6674 to 2.062
Carbohydrate intake Normal	10		136		1	
High and very high carbohydrate intake	26		308		1.148	0.5386 to 2.447

**Table 3 children-09-01510-t003:** Distribution of the colonization by *Candida* spp. according to age.

*Candida* spp.	Age (Years)			
	3	4	5	6
Presence	0	20	21	11
Absence	39	270	309	73

## Data Availability

The data presented in this study are available on request from the corresponding author. The data are not publicly available due to the fact that they are personal data of each patient.

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
