# Peer review of "Frequency of Candida spp. in the Oral Cavity of Asymptomatic Preschool Mexican Children and Its Association with Nutritional Status"

_children, 2022, doi:10.3390/children9101510_

Round 1

Reviewer 1 Report

This study looking at the association between presence of Candida and nutritional status in population of Mexican children is much needed study looking at the wider microbiome in non-Western cultures. The study used a large population with a good distribution of nutrirional status and combined culture plating to identify presense or absence of Candida and metadata collected from parent surveys looking specifically at simple CHO consumption. I could not find the survey questions in the supplementary data. Did you look at complex CHO?  the % of overweight and obese children suggest that this may be something to investigate as well. However, my significant issue is with the methodology, while the methodology is well described and appropriate for a culture based study, these types of study are no longer relevant when genomic identification is readily available. This study would have scientifica impact if Candida was identified using 16s rRNA, this would also provide a much more comprehenisve view of the mycobiome in these children. I would strongly advise this is done on representiative sample (smaller population) and by combining this with the larger culture study it would have contemporary impact. Finally please rephrase the use of toilet in line 82 (this is not contemporary medical language). 

Author Response

Answers to Reviewer 1 concerns:

This study looking at the association between presence of Candida and nutritional status in population of Mexican children is much needed study looking at the wider microbiome in non-Western cultures. The study used a large population with a good distribution of nutrirional status and combined culture plating to identify presense or absence of Candida and metadata collected from parent surveys looking specifically at simple CHO consumption.

We are thankful for the time and effort you have invested in the revision of our manuscript. All your suggestions have enriched our work. In the manuscript, the additions are highlighted in yellow. Please find our answers to your valuable recommendations; we hope that we have addressed all your concerns.

  1. I could not find the survey questions in the supplementary data. Did you look at complex CHO? the % of overweight and obese children suggest that this may be something to investigate as well.

Answer: Thank you for your kind suggestion. The food survey questions have been attached to the supplementary material.

  1. However, my significant issue is with the methodology, while the methodology is well described and appropriate for a culture-based study, these types of study are no longer relevant when genomic identification is readily available. This study would have scientifica impact if Candida was identified using 16s rRNA, this would also provide a much more comprehenisve view of the mycobiome in these children. I would strongly advise this is done on representiative sample (smaller population) and by combining this with the larger culture study it would have contemporary impact.

Answer: Thank you for your kind suggestion. The study deals with the relationship between the nutritional status of children from 3 to 6 years of age and its association with the presence of Candida spp. in the mouth, therefore, was to determine whether or not yeast was present in each of the children whose mouths were sampled, the study was not intended to identify the species in a molecular way but in a phenotypic way and therefore the decision was made to identify the most common species with the CHROMagar-Candida, which identifies C. albicans, C. tropicalis, C. krusei, C. glabrata and C. parapsilosis. There is also the inconvenience of doing PCR at this time because some of the strains are no longer viable in their growth.

  1. Finally please rephrase the use of toilet in line 82 (this is not contemporary medical language). 

Answer: This is an excellent observation. The sentence was corrected “not having toilet orally” for “not having oral or tooth washing”

Reviewer 2 Report

ABSTRACT: slight modification of language especially line 27 and line 30.  

INTRODUCTION: Clarity needed for lines 66-69 and lines 73-75

Line 77 - How are the children "marginalized" please briefly explain

Line 87 - The sentence needs to be restructured to enhance clarity. Translation needs improvement

METHODOLOGY: confirm presence of yeast by Gram Stain? This is inadequate. The experimental design need to be improved or properly explained. More information needed. Were the samples only collected from the cheek?

RESULTS: Explain or state what the following nutritional status mean in this study - Normal nutritional status, undernutrition, overweight and obese

Author Response

Answers to Reviewer 2 concerns: We appreciate the time and effort you have invested in the revision of our manuscript. Indeed, all your suggestions have improved the quality of our manuscript. In the main text, the additions are highlighted in yellow. We hope that we have correctly addressed all your concerns. 1. Abstract: slight modification of language especially line 27 and line 30. Answer: Thank you for your valuable comments. The pertinent modifications were made, which are highlighted in yellow in the paper. 2. Line 77 - How are the children "marginalized" please briefly explain
Answer: Thank you for your kind suggestion. It refers to the fact that it is possible to identify the territorial disparities existing in the country at a given time, a quality that has given it a relevant value as an analytical and operational tool for the definition and targeting of public policies, focused on reducing the socioeconomic deprivation of the Mexican population. Thus, we have added the word social before marginalized.
3. Line 87 - The sentence needs to be restructured to enhance clarity. Translation needs improvement. Answer: Thank you for your suggestion. The wording of the aforementioned sentence was corrected and highlighted in yellow in the paper. 4. Methodology: confirm presence of yeast by Gram Stain? This is inadequate. Answer: Thank you for the opportunity to clarify this sentence. It was corrected in lines 107 to 110, where the correct technique used is mentioned. You can find it highlighted in yellow in the paper. 5. The experimental design needs to be improved or properly explained. More information needed. Answer: Thank you for your suggestion. The experimental design within the methodology was improved in section 2.2. the paper. 6. Were the samples only collected from the cheek?
Answer: Thank you for your valuable comments. As mentioned in section 2.2, line 98, samples were taken from the different areas of the mouth (cheeks, the dorsum of the tongue, floor of the mouth and palate). 7. Results: Explain or state what the following nutritional status mean in this study - Normal nutritional status, undernutrition, overweight and obese Answer: Thank you for these kind observations. This suggestion was addressed in the discussion section, which is highlighted in yellow in the paper.

Reviewer 3 Report

Very good paper with good methodology

Author Response

Answers to Reviewer 3 concerns: Very good paper with good methodology We appreciate the time and effort you have invested in the revision of our manuscript. Indeed, all your suggestions have improved the quality of our manuscript. In the main text, the additions are highlighted in yellow. We hope that we have correctly addressed all your concerns.
Answer: Thank you for your motivating words, we hope that it has been of interest to you and we hope that it will be useful for various researchers and pediatricians.

Round 2

Reviewer 1 Report

The minor correction is to line 82 The sentence was corrected “not having toilet orally” for “not having oral or tooth washing”

It should read "not using mouth rinses or tooth brushing".

Author Response

Dear reviewer, we thank you very much for taking the time to review our manuscript and let you know that the last correction you suggested has been made.
In line 82 where it said: "not having oral or tooth washing" was changed by your suggestion  "not using mouth rinses or tooth brushing".

Thank you very much
